# Relative Behavioral Attributes: Filling the Gap between Symbolic Goal Specification and Reward Learning from Human Preferences

**Lin Guan, Karthik Valmeekam, Subbarao Kambhampati**
School of Computing & AI
Arizona State University
Tempe, AZ 85281, USA
{lguan9,kvalmeek,rao}@asu.edu

## Abstract

Generating complex behaviors that satisfy the preferences of non-expert users is a crucial requirement for AI agents. Interactive reward learning from trajectory comparisons (a.k.a. RLHF) is one way to allow non-expert users to convey complex objectives by expressing preferences over short clips of agent behaviors. Even though this parametric method can encode complex tacit knowledge present in the underlying tasks, it implicitly assumes that the human is unable to provide richer feedback than binary preference labels, leading to intolerably high feedback complexity and poor user experience. While providing a detailed symbolic closed-form specification of the objectives might be tempting, it is not always feasible even for an expert user. However, in most cases, humans are aware of how the agent should change its behavior along meaningful axes to fulfill their underlying purpose, even if they are not able to fully specify task objectives symbolically. Using this as motivation, we introduce the notion of *Relative Behavioral Attributes*, which allows the users to tweak the agent behavior through symbolic concepts (e.g., increasing the softness or speed of agents' movement). We propose two practical methods that can learn to model any kind of behavioral attributes from ordered behavior clips. We demonstrate the effectiveness of our methods on four tasks with nine different behavioral attributes, showing that once the attributes are learned, end users can produce desirable agent behaviors relatively effortlessly, by providing feedback just around ten times. This is over an order of magnitude less than that required by the popular learning-from-human-preferences baselines. The supplementary video and source code are available at: https://guansuns.github.io/pages/rba.

## 1 Introduction

A central problem in building versatile autonomous agents is how to specify and customize agent behaviors. Two representative ways to specify tasks include manual specification of reward functions, and reward learning from trajectory comparisons. In the former, the user needs to provide an exact description of the objective as a suitable reward function to be used by the agent. This is often only feasible when the specification can be done at a high level, e.g. in symbolic terms (Russell & Norvig, 2003) or by providing a symbolic reward machine or domain model (Yang et al., 2018; Illanes et al., 2020; Guan et al., 2022; Icarte et al., 2022), or by giving a natural language instruction (Mahmoudieh et al., 2022). Instead of requiring users to give precise task descriptions, reward learning from trajectory comparisons (a.k.a. reinforcement learning from human feedback or RLHF) learns a reward function from human preference labels over pairs of behavior clips (Wilson et al., 2012; Christiano et al., 2017; Ouyang et al., 2022; Bai et al., 2022a), or from numerical ratings on trajectory segments (Knox & Stone, 2009; MacGlashan et al., 2017; Warnell et al., 2018; Guan et al., 2021; Abramson et al., 2022) or from rankings over a set of behaviors (Brown et al., 2019).

To illustrate the distinct characteristics of the two objective specification methods, let us consider a Humanoid control task in a benchmark DeepMind Control Suite (Tunyasuvunakool et al., 2020).

The default *symbolic* reward function in the benchmark is a carefully-designed linear combination of multiple explicit factors such as moving speed and control force. By optimizing the reward, the agent will learn to run at the specified speed but in an unnatural way. However, to further specify the motion styles of the robot (e.g., to define a human-like walking style), non-expert users may find it hard to express such objectives symbolically since this involves *tacit* motion knowledge. Similarly, in natural language processing tasks like dialogue (Ouyang et al., 2022) and summarization (Stiennon et al., 2020), it can be extremely challenging to construct reward functions purely with explicit concepts. Hence, for tacit-knowledge tasks, people usually resort to reward specification via pairwise trajectory comparisons and learn a parametric reward function (e.g., parameterized by deep neural networks). While pairwise comparisons are general enough to work for any setting, due to the limited information a binary label can carry, they are also an impoverished way for humans to communicate their preferences. Thus, treating every task at hand as a tacit-knowledge one and limiting reward specification to binary comparisons can be unnecessarily inefficient. Moreover, since the internal representations learned within neural networks are typically inscrutable, it's unclear how a learned reward model can be reused to serve multiple users and fulfill a variety of goals.

In short, symbolic goal specification is more straightforward and intuitive to use, but it offers limited expressiveness, making it more suitable for explicit-knowledge tasks. Reward learning from trajectory comparisons, in contrast, offers better expressiveness, but it is more costly and less intuitive to use. However, in many real-world scenarios, user objectives are neither completely explicit nor purely tacit. In other words, although human users may not be able to construct a reward function symbolically, they still have some idea of how the AI agent can change its behavior along certain meaningful axes of variation to better serve the users. In the above Humanoid example, even though the users can not fully define the motion style in a closed-form manner, they may still be able to express their preferences in terms of some nameable concepts that describe certain properties in the agent behavior. For instance, the users of a household humanoid robot may want it to not only walk in a natural way but also walk *more softly* and take *smaller steps* at night when people are sleeping.

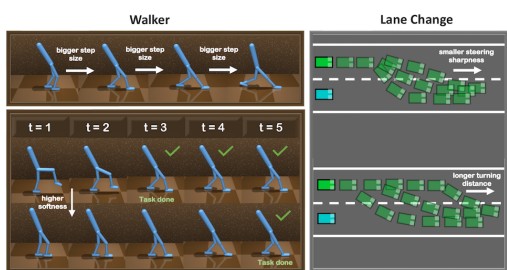

Figure 1: Visualizing behavioral attributes of Walker and Lane-Change. The behavioral attributes of other domains are shown in Fig. 3 in Appendix A.

Motivated by the observations above, we introduce the notion of *Relative Behavioral Attributes* (RBA) to capture the relative strength of some properties' presence in the agent's behavior. We aim to learn an attribute-parameterized reward function that can encode complex task knowledge while allowing end users to freely increase or decrease the strength of certain behavioral attributes such as "step size" and "movement softness" (Fig. 1). This new paradigm of reward specification is intended to bring the best of symbolic and pairwise trajectory comparison approaches by allowing for reward feedback in terms of conceptual attributes. The benefits of explicitly modeling RBAs are obvious: (a) it offers a semantically richer and more natural mode of communication, such that the hypothesis space of user intents or preferences can be greatly reduced each time the system receives attribute-level feedback from the user, thereby improving the overall feedback complexity significantly; (b) since humans communicate at the level of symbols and concepts, the learned RBAs can be used as shared vocabulary between humans and inscrutable models (Kambhampati et al., 2022), and more importantly, such shared vocabulary can be *reused* to serve any future users and support a diverse set of objectives.

The concept of relative attributes has been introduced earlier in other areas like computer vision (Parikh & Grauman, 2011) and recommender systems (Goenka et al., 2022). Though they share some similarities in motivations and definition to our relative behavioral attribute, the key difference is that relative attributes there only capture properties present in a single static image, while RBAs aim to capture properties over trajectories of behavior, wherein an attribute may be associated with either static features at a certain timestep (e.g., step size of a walking agent) or temporally-extended features spanning multiple steps (e.g., the softness of movement). Additionally, we need to consider

how to connect the captured attributes to a valid reward function. In this work, we propose two generic data-driven methods to encode RBAs within a reward function. On four domains with nine attributes in total, we show that our methods can learn to accurately capture the attributes from roughly 200 labelled behavior clips. Once the attribute-parameterized reward is learned, any end user can produce diverse agent behaviors with relative ease, by providing feedback just around 10 times. This offers a significant advantage over the learning-from-human-preferences baseline (Christiano et al., 2017) that requires hundreds of binary labels from the user per task.

## 2 RELATED WORK

In general, direct reward specification is the most commonly employed approach. It is used in almost all simulators with engineered reward functions and in industrial applications such as self-driving vehicle control systems. Yet, there are still some challenges associated with it, such as extensive requirement for sensory instrumentation, incomplete (sub)goal specification (Guan et al., 2022), reward exploitation (Lee et al., 2021) and the expressiveness issue mentioned in the previous section. These impediments motivate the idea of learning reward from states or trajectory comparisons (Christiano et al., 2017; Warnell et al., 2018; Zhang et al., 2019; Lee et al., 2021), which leverages the exceptional representational capacity of neural networks but tends to suffer from high data complexity. Another alternative to symbolic reward specification is imitation learning or inverse reinforcement learning (Schaal, 1996; Ng et al., 2000; Abbeel & Ng, 2004). However, this is usually infeasible for non-expert end-users, as it requires the user to have sufficient knowledge and proper hardware setup in order to teleoperate the agent. Even if a pre-collected large-scale behavior dataset is provided to the user, the process of finding the target behavior trace(s) from the large dataset can still be frustrating.

The introduction of RBAs aligns with the broader intent of building symbolic interfaces as a middle layer for humans to communicate effectively with the agent (Bobu et al., 2021; Guan et al., 2021; 2022; Silver et al., 2022; Zhang et al., 2022; Bucker et al., 2022; Cui et al., 2023) or for the agent to explain to humans (Kim et al., 2018; Sreedharan et al., 2022; Kambhampati et al., 2022). Lee et al. (2020) utilize relative-attribute information in robot skill learning, but their GAN-based formulation is restricted to static visual attributes and is not applicable to temporally-extended concepts.

This paper adopts a similar setup to works that learn diverse skills or motion styles from large-scale offline behavior datasets or demonstrations (Lee & Popović, 2010; Wang et al., 2017; Zhou & Dragan, 2018; Peng et al., 2018b; Luo et al., 2020; Chebotar et al., 2021; Peng et al., 2021). These works emphasize on modeling a variety of reusable motor skills by learning a low-level controller conditioned on skill latent codes. Since the latent codes are inscrutable to humans, for each new task, the user must specify the desirable agent behavior by constructing an engineered *symbolic reward* and use it to train a separate high-level policy that controls the low-level controller. Our methods are complemented by existing diverse-skill learning methods because skill priors (i.e., pre-trained low-level controllers) allow us to optimize the behavioral reward more efficiently. More recently, there have been works in diffusion-based text-to-motion animation generation (Tevet et al., 2022; Guo et al., 2022). They are similar to this work in the sense that we both allow humans to control the agent behavior through explicit concepts. However, they do not support fine-grained control over the strength of individual behavioral attributes, and their works are not applicable to physics-based character control.

## 3 PROBLEM SETUP

**Personalizing agent behaviors at skill level.** We assume the users are interested in customizing agent behaviors at the level of skills (e.g., making a lane change, taking a step, picking up an object). A skill is a solution (i.e., policy) to an episodic task in an indefinite-horizon discounted MDP $\mathcal{M} = \langle S, A, R, P, \gamma \rangle$, where $S$ is the set of states, $A$ is the set of actions, $R : S \times A \to \mathbb{R}$ is the provided reward function, $P : S \times A \times S \to [0, 1]$ is the transition function and $\gamma$ is the discount factor. An episode terminates whenever the agent reaches an absorbing terminal state or exceeds a fixed number of time steps $T$. An agent $M$ is supposed to find a policy ($\pi : S \to A$) that maximizes the expected return $E[\sum_{t=0}^{T} \gamma^t r_t], r \in \mathbb{R}$. When executing a policy, the agent is initialized to some initial state $s_0 \in S$ at the beginning of each episode; and then at each succeeding time step $t$, the

agent interacts with the environment $\mathcal{E}$ by taking an action $a_t \in A$ and receiving the next state $s_{t+1} \in S$. In order to personalize agent behavior, rather than assuming that the reward function $R$ is supplied by the environment, we require the reward function to be specified by the end user.

Given a behavior clip or a trajectory $\tau = \{(s_0, a_0), ..., (s_l, a_l)\}$, where $l$ is the length of the trajectory, a relative behavioral attribute $\alpha \in \mathcal{A}$ captures the strength of the presence of a certain property exhibited in $\tau$. It assumes that a mapping $\zeta$ can be established to map any $(\alpha, \tau)$ pair to a real value that reflects the relative strength of $\alpha$ in $\tau$. In this work, our goal is to construct a reward function that allows end users to iteratively adjust attribute strengths presented in the agent's behavior.

**Learning to model RBAs from offline behavior datasets.** RBAs are essentially semantically capturing different ways to carry out a task. Ideally, the training data for the reward function should contain clips of diverse skill behaviors. Although we used synthetic data in this study, there are many publicly accessible behavior corpora, such as the Waymo Open Dataset for autonomous driving tasks (Ettinger et al., 2021) and large-scale motion clips data for character control (Peng et al., 2018a; 2022). Note that we do not expect the offline dataset to exhaustively cover all possible skill behaviors, as the reward function should generalize to unseen configurations. Also, considering that action information is not always available, to be more general, we assume the training dataset $\mathcal{D}$ only consists of state-only trajectories. Last but not least, as a common setting, we assume the embodiment of the agent is not significantly different from that in $\mathcal{D}$.

**Reward learning from trajectory comparisons.** Methods for reward learning from trajectory comparisons construct a reward function by learning to infer the user's latent preference from a set of ranked trajectories $\{(\tau^1, \tau^2)\}$. Typically, the user preference is modelled according to the Bradley-Terry model (Bradley & Terry, 1952):

$$P\left[\tau^1 \succ \tau^2\right] = \frac{\exp \sum_t r\left(s_t^1, [a_t^1]\right)}{\sum_{i \in \{1,2\}} \exp \sum_t r\left(s_t^i, [a_t^i]\right)}, \tag{1}$$

where $\tau^1 \succ \tau^2$ denotes the event that the user prefers $\tau^1$ over $\tau^2$, $r$ is a parametric learnable reward function, and $[a_t^i]$ means the action input is optional. A cross-entropy loss is typically used for optimization:

$$Loss_p(r) = - \sum_{(\tau^1, \tau^2, y) \in \mathcal{D}_p} y(1) \log P\left[\tau^1 \succ \tau^2\right] + y(2) \log P\left[\tau^2 \succ \tau^1\right], \tag{2}$$

where $y \in \{(1,0), (0,1), (0.5, 0.5)\}$ are possible preference labels indicating $\tau^1 \succ \tau^2$, $\tau^2 \succ \tau^1$, or $\tau^1$ and $\tau^2$ are equally preferred, respectively. The set of preference labels $\mathcal{D}_p$ can be collected beforehand (Brown et al., 2019) or through active queries to the user in an online manner (Wilson et al., 2012; Christiano et al., 2017). The latter is often referred to as preference-based reinforcement learning, or PbRL for short. Most PbRL methods build upon the Bradley-Terry model but differ in the query strategies and how the network is initialized (Lee et al., 2021; Park et al., 2022; III & Sadigh, 2022; Ren et al., 2022).

## 4 METHODOLOGY

### 4.1 PERSONALIZING AGENT BEHAVIOR VIA RELATIVE BEHAVIORAL ATTRIBUTES

Our framework involves two phases, namely, learning an attribute parameterized reward function and interacting with the end user.

**Learning attribute parameterized reward function (no interaction with the end user).** This phase is supposed to learn a reward function that *internally* learns a family of rewards that correspond to behaviors with diverse attribute strengths. By varying the input attribute configurations, end users will be able to find the preferred "reward member" and obtain desired behaviors. In Section 4.2 and Section 4.3, we will present two methods that can learn such a reward function given a subset of labelled trajectories from an offline behavior dataset $\mathcal{D}$. We assume that the *agent builders* are the ones who provide the training labels (e.g., the engineers of autonomous vehicles, and the developer of virtual characters). In some extreme cases, if a novel concept has to be learned, the training labels may also come from the end user. Also note that, this step *can be skipped* if the RBAs have already been captured by a learned reward function.

**Supporting end users in the loop.** Once an attribute parameterized reward function is learned, any incoming users can leverage it to personalize the agent behavior through multiple rounds of query. In each round of interaction, the agent presents the user a trajectory, sampled according to the policy that optimizes the current reward function. Then the user provides a feedback on whether the current behavior is desirable; and if not satisfied, the user can express the intent to increase or decrease the strength of certain attributes. The concept-level feedback is a set of attribute-feedback pairs $\{(\alpha, h)\}$, where $\alpha$ is the attribute of interest, and $h$ is a binary value indicating whether the user wants to increase or decrease $\alpha$'s strength. In this work, we consider two types of attribute representation, namely the *index* of $\alpha$ in a list of known attributes or a *natural-language description* of $\alpha$. Upon collecting feedback from the user, the agent will adjust the reward function and update the corresponding policy. This human-agent interaction process repeats until the user is satisfied with the latest agent behavior.

In the next two sections, we will elaborate on two candidate architectures for the attribute parameterized reward function. We note that both architectures support the same type of user interaction as outlined above.

### 4.2 METHOD 1: MODELING BEHAVIORAL ATTRIBUTES BY ESTABLISHING GLOBAL RANKINGS

The learning process of Method 1 consists of two steps. In the first step, we learn an attribute strength estimator $\zeta_\sigma$ (parameterized by $\sigma$) that can map any given attribute $\alpha$ and trajectory $\tau$ to a real-valued score that measures the relative strength of attribute $\alpha$ in $\tau$. Here, $\zeta_\sigma$ is essentially establishing a *global ranking* among all possible behaviors according to any given attribute $\alpha$. Hence, we denote this method as **RBA-Global**. Then in the second step, given a finite set of attributes $\mathcal{A}$, we learn a dense reward function $r_\theta(s | v_t = \langle v_t^{\alpha_1}, ..., v_t^{\alpha_k} \rangle)$, where $v_t$ is called the target attribute-score vector, $v_t^{\alpha_i}$ is the target strength of attribute $\alpha_i$, and $k = |\mathcal{A}|$ is the total number of attributes. $r_\theta$ is expected to satisfy that, when $r_\theta(\cdot | v_t)$ is optimized, the agent is able to sample a trajectory $\tau'$, such that for any $\alpha_i \in \mathcal{A}$, we have $\zeta_\sigma(\tau', \alpha_i) \simeq v_t^{\alpha_i}$. Hence, with a learned reward function $r_\theta$, the agent can produce diverse behaviors by varying the input target attribute-score vector $v_t$. As an example, let us consider the humanoid household robot domain with two attributes $\{\alpha_0 : \text{"softness of movement"}, \alpha_1 : \text{"step size"}\}$. By setting the target attribute vector $v_t$ to be $\langle v_t^{\alpha_0} = -1.5, v_t^{\alpha_1} = 2.0 \rangle$, we expect the agent that optimizes $r_\theta$ to be able to produce a trajectory $\tau'$ with a "softness" score $\zeta_\sigma(\tau', \alpha_0)$ approximately equal to $-1.5$. A graphical illustration of the overall architecture can be found in Fig. 4 in Appendix A. In the remainder of this section, we will explain how $\zeta_\sigma$ and $r_\theta$ can be learned from the offline behavior dataset $\mathcal{D}$.

The problem of learning an attribute strength estimator $\zeta_\sigma$ is essentially a learning-to-rank problem. Specifically, we assume we are given (normally by the agent designers rather than the end users) a set of state-only trajectories $\{\tau\}$ and their orderings according to different attributes $\{(\tau^0 \succ \tau^1 \succ ... \succ \tau^N | \alpha)\}$, or a set of ranked trajectory pairs $\mathcal{D}_l = \{(\tau^i \succ \tau^j | \alpha)\}$, where $\alpha \in \mathcal{A}$ is one of the attributes in the domain, $(\tau^i \succ \tau^j | \alpha)$ represents the event that the attribute $\alpha$ has a stronger presence in trajectory $\tau^i$ than that in trajectory $\tau^j$, and $N \leq |\mathcal{D}|$ is the length of the ranked trajectory sequence. We propose to employ a modified state-only version of Bradley-Terry model (Eq. 1), in which rather than assuming that the ranking is governed by the latent user preferences, we assume the ranking is determined by the given attribute $\alpha$:

$$P_\sigma \left[ \tau^1 \succ \tau^2 \,\middle|\, \alpha \right] = \frac{\exp \sum_t f_\sigma \left( [s_t^1, e_\alpha] \right)}{\sum_{i \in \{1,2\}} \exp \sum_t f_\sigma \left( [s_t^i, e_\alpha] \right)}, \tag{3}$$

where $f_\sigma$ is an attribute conditioned ranking function with parameters $\sigma$, $[\cdot, \cdot]$ is the vector concatenation operation, and $e_\alpha$ is the embedding of attribute $\alpha$. The strength of any attribute $\alpha$ in a trajectory $\tau$ is given as $\zeta_\sigma(\tau, \alpha) = \sum_{s \in \tau} f_\sigma \left( [s, e_\alpha] \right)$. Recall that we consider two types of attribute representation, namely attribute index and natural language description. Accordingly, $e_\alpha$ can either be a one-hot vector or a sentence embedding generated by any pretrained natural language sentence encoder like Sentence-BERT (Reimers & Gurevych, 2019). Since different behaviors may result in trajectories of varying lengths, in Eq. 3 we do not require the two trajectories to have the same size. Given the training dataset $\mathcal{D}_l$, $f_\sigma$ can be trained via a cross-entropy loss similar to the one in Eq. 2. For numerical stability, in practice, we also clip the values of $\sum_t f_\sigma \left( [s_t^i, e_\alpha] \right)$ (we used $[-20, 20]$ for all the attributes in our experiment). With a learned attribute strength estimator $\zeta_\sigma$ and a finite

set of $k$ attributes, the agent behavior in any trajectory $\tau$ can be characterized by an attribute vector $v(\tau) = \langle \zeta_\sigma(\tau, \alpha_1), ..., \zeta_\sigma(\tau, \alpha_k) \rangle$.

The problem of learning $r_\theta([s, v_t])$ can also be cast to a learning-to-rank or preference modeling problem, wherein the reward function is supposed to give higher cumulative rewards to trajectories that have attribute strengths closer to the targets in $v_t$. Specifically, given $v_t$, for any two trajectories $\tau_i$ and $\tau_j$ from the offline behavior dataset $\mathcal{D}$, the preference label $l_p(\tau^i, \tau^j, v_t)$ is given as:

$$l_p(\tau^i, \tau^j, v_t) = \begin{cases} \tau^i \succ \tau^j & \text{if } \|v(\tau^i) - v_t\|_2 < \|v(\tau^j) - v_t\|_2 - \xi_r \\ \tau^i \prec \tau^j & \text{if } \|v(\tau^i) - v_t\|_2 > \|v(\tau^j) - v_t\|_2 + \xi_r \\ \text{no ordering} & \text{otherwise,} \end{cases} \quad (4)$$

where $\xi_r$ is a small slack variable. To train $r_\theta$, we can randomly sample a set of triplets from $\mathcal{D}$, namely $\{(\tau^0, \tau^1, \tau^2)\}$. By treating $\tau^0$ as the target behavior, we can generate a set of training labels $\{l_p(\tau^1, \tau^2, v_t = v(\tau^0))\}$ for $r_\theta$. In short, $r_\theta$ can be trained as a standard state-only preference-based reward function according to Eq. 1 and Eq. 2 but with preference labels given by the extracted attribute strengths (i.e., by $\zeta_\sigma$). Note that, unlike the training of $\zeta_\sigma$, the training of $r_\theta$ does not require any human-provided labels.

Once a reward function $r_\theta$ is learned, the end user can use it to specify agent behavior by simply tuning the attribute strength values in the input target attribute vector $v_t$. In our realization, we implement the process of finding the target attribute score as a process of performing binary search in real space (details can be found in Appendix A.3). The process of personalizing agent behavior through $r_\theta$ is highly intuitive because $r_\theta$ handles the complex tacit parts of the problem internally (e.g., how to walk naturally and realistically with the constraints of softness and step size) and only relies on the end user to set the explicit parts. Additional discussion on alternative ways to leverage $\zeta_\sigma$ without learning a reward function can be found in Appendix A.2.

### 4.3 METHOD 2: MODELING BEHAVIORAL ATTRIBUTES BY CAPTURING MINIMALLY VIABLE LOCAL CHANGES

One limitation of RBA-Global is that, it requires the total number of encoded attributes to be finite because the size of the target attribute vector $v_t$ grows with the number of attributes. This may limit the scalability of RBA-Global. In this section, we will introduce a more extensible method that can potentially encode an arbitrary number of attributes. The key motivation is that in RBA-Global, the user never directly manipulates the attribute scores. Instead, we can skip the explicit modeling of attribute strength (i.e., learning $\zeta_\sigma$) and directly learn a *behavior-editing* reward function.

Specifically, given a trajectory $\tau_c$ and the corresponding human feedback $(\alpha, h)$, our goal is to construct a reward function $r_\theta(\cdot | \alpha, h, \tau_c)$ that gives higher cumulative rewards to trajectories that have some minimal but noticeable change in $\alpha$ in the direction specified by $h$ while keeping other unmentioned attributes unchanged (or minimally changed). We refer to such minimal but noticeable changes as *minimally viable local changes*, and the queried trajectory $\tau_c$ as the *anchor trajectory*. Accordingly, we denote this method as **RBA-Local**.

To learn $r_\theta$, we assume we are provided (again, normally by the agent builder and not the end user) a set of trajectory pairs $D_l = \{(\tau_c, \tau_t, \alpha, h)\}$, where $\tau_t$ is a trajectory that reflects some minimally viable local changes to the anchor trajectory $\tau_c$ in terms of the attribute $\alpha$ and the direction $h$. $r_\theta$ is trained to prefer $\tau_t$ over other negative samples (such as trajectories that make excessive changes to $\tau_c$, or trajectories that are not significantly different from $\tau_c$, or trajectories that make changes to unspecified attributes). In practice, we select the negative samples by randomly sampling from the behavior dataset $\mathcal{D}$. Again, $r_\theta$ can be formulated as a modified Bradley-Terry model:

$$P_\theta[\tau_t \succ \tau_n | \alpha, h, \tau_c] = \frac{\exp \sum_{s \in \tau_t} r_\theta([s, e_\alpha, h, \phi(\tau_c)])}{\exp \sum_{s \in \tau_n} r_\theta([s, e_\alpha, h, \phi(\tau_c)]) + \exp \sum_{s \in \tau_t} r_\theta([s, e_\alpha, h, \phi(\tau_c)])}, \quad (5)$$

where $\tau_n$ is a negative sample (i.e., trajectory), $[\cdot]$ is the vector concatenation operation, $e_\alpha$ as in RBA-Global can be either an one-hot representation of attribute $\alpha$ or a sentence embedding output by Sentence-BERT, and $\phi(\tau_c)$ is a sequence encoder (e.g., an LSTM (Hochreiter & Schmidhuber, 1997)) that encodes the anchor trajectory $\tau_c$ to a compact latent representation. Note that $\phi(\cdot)$ is a sub-module of $r_\theta$ and it's jointly optimized with $r_\theta$. Since $r_\theta$ is essentially a preference-based reward function, it can be optimized by employing a cross-entropy loss as in the one in Eq. 2. We

note that our computational framework is similar to the Prompt-DT (Xu et al., 2022) in the sense that we both take a reference trajectory as a "prompt" to the model to obtain conditioned outputs. But instead of trying to replicate the behavior in the "prompt" trajectory as in Prompt-DT, our reward function learns to modify the prompt in a controlled way. In a more recent work (Liu et al., 2023), similar ideas have been shown to be effective in refining language model outputs through sequences of local changes informed by human feedback.

Compared to RBA-Global (Sec. 4.2) which requires a full specification of all the attributes' strengths, the reward function in RBA-Local only takes one attribute as input at a time. This design is appealing as it offers better scalability and it affords the development of a big universal behavioral concept "encoder". Nevertheless, RBA-Local still has the following shortcomings: (a) it is less efficient than RBA-Global in searching for the target behavior because it can only make minimally viable changes to the presented behavior; (b) The training data for RBA-Local is harder to collect. Unlike RBA-Global, where any random subset of trajectories exhibiting distinct behaviors can be used as the training samples, in RBA-Local, the agent builder must carefully pick pairs of trajectories that reflect local changes. Also, the judgement of how much variation constitutes a minimally viable change can be fairly subjective.

## 5 Empirical Evaluation

As a proof of concept, we demonstrate the effectiveness of our methods in a diverse set of four domains with nine behavioral attributes that are depicted in Fig. 1 and Fig. 3:

**Walker.** This environment corresponds to a scenario that involves a 2-legged home-service robot walking around the house to perform household tasks. The users may want the robot to walk more softly at night. This environment is also related to physical character control scenarios wherein we want the character to move in a sneaky way. Two attributes are considered here: (a) step size; (b) softness of movement.

**Manipulator.** We consider a virtual character control scenario wherein we want a simulated arm to mimic the ways of a human lifting objects. When humans, especially elders and children, are lifting heavy objects, their movement can be unstable. Hence, we consider two attributes: (a) moving speed of the arm; (b) instability of the movement.

**Lane Change.** We consider a driving scenario wherein the rider would want to change the lane-changing behavior of the cab to get a more pleasant experience. Two attributes are used for evaluation: (a) the sharpness of steering: this attribute corresponds to how sharp a turn the agent makes while changing lanes; (b) distance to the following vehicle: this attribute is about the distance between our agent and the following car at the moment when our agent starts making the lane change.

**Snake Concertina.** In this task, the agent is supposed to control a virtual snake to imitate diverse concertina styles of a real snake's locomotion. There are three relevant attributes in concertina locomotion: (a) width of the bend (i.e., the maximal width that the snake occupies); (b) compression (i.e., how much the snake's body is compressed when it is moving); (c) speed of movement.

More details of the evaluation domains can be found in Appendix A.1. In our experiments, all the behavior clips are generated either by hard-coded motions or by using reinforcement learning with sophisticated reward designs and hard-coded constraints.

| Method | Lane-Change | | Manipulator | | Snake | | Walker | |
|---|---|---|---|---|---|---|---|---|
| | SR | AF (std) | SR | AF (std) | SR | AF (std) | SR | AF (std) |
| RBA-Global | 0.95 | 3.95 *(2.43)* | 1.0 | 2.8 *(1.21)* | **0.85** | **4.17 *(1.85)*** | 1.0 | **3.75 *(1.47)*** |
| RBA-Global-L | **1.0** | **3.05 *(2.06)*** | 1.0 | **2.5 *(1.32)*** | 0.8 | 6.38 *(5.03)* | 0.95 | 3.78 *(2.25)* |
| RBA-Local | 0.7 | 7.07 *(3.49)* | 0.7 | 12.23 *(5.75)* | 0.55 | 6.45 *(3.82)* | 0.95 | 5.47 *(3.52)* |
| RBA-Local-L | 0.9 | 5.78 *(4.04)* | 0.8 | 10.75 *(6.35)* | 0.6 | 4.17 *(2.11)* | 0.9 | 5.16 *(3.08)* |
| PbRL | 1.0 | 162.3 *(184)* | 0.6 | 159.5 *(188.87)* | 0.05 | N/A | 1.0 | 84.6 *(79.87)* |

Table 1: SR - Success Rate; AF - Average Feedback (when success); L - Language

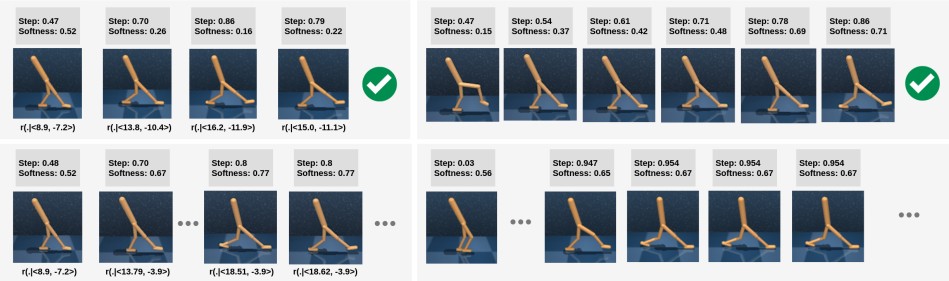

Figure 2: A step-by-step visualization of the interaction process showing how the agent's behaviors change according to sequences of user's feedback in the Walker domain. The two sequences on the left showcase RBA-Global and the sequences on the right showcase RBA-Local. The upper and bottom rows represent the success and failure cases respectively. The attribute strength scores above each image frame are given by hard-coded proxy measures, which are in the range of $[0, 1]$. For RBA-Global, we also show the corresponding reward function and input $v_t = \langle \text{target step size}, \text{target softness} \rangle$. A more detailed visualization is presented in the supplementary video.

## 5.1 BASELINE AND RESULTS

For comparisons, we use the PbRL algorithm proposed in (Christiano et al., 2017; Lee et al., 2021), which learns a reward function from human preference labels collected by making active queries. Considering that our methods assume the additional access to the offline behavior dataset $\mathcal{D}$, we made a couple of modifications to make PbRL into a stronger baseline. The most important one is that, to optimize the most recent reward function and update the agent's behavior after each query, rather than applying reinforcement learning, we use the policies extracted from $\mathcal{D}$. Specifically, we stochastically sample a large set of rollouts by executing the policies we used to synthesize dataset $\mathcal{D}$, and the rollout with the highest cumulative rewards is set as the agent's latest behavior. In practice, these policies can also be unsupervisedly learned from $\mathcal{D}$. This is identical to the use of skill priors as in (Peng et al., 2018b; Pertsch et al., 2020; Luo et al., 2020; Peng et al., 2022), which first learns a diverse set of natural and plausible motions from offline behavior datasets to reduce online computation. Besides, for PbRL, we also experimented with different query strategies and considered reusing previously trained reward models. For the sake of simplicity, we only report the best PbRL performance achieved at different setups.

As the evaluation metric, we count the number of human feedbacks (i.e., the binary preference labels in PbRL and the attribute-level feedback in our methods) needed to produce the target behavior. For each domain, we randomly sample 20 behavior configurations as targets, and the selected targets were unseen in the behavior dataset $\mathcal{D}$. A trial is considered as a success if the generated agent behavior has a ground truth attribute strength or proxy score that falls within a certain range of the target value. We use a threshold that roughly divides the strength of each attribute into five to ten buckets. A trial is deemed unsuccessful if the agent fails to produce the target behavior within a user-affordable number of feedbacks (we used 500 in our evaluation). The results are shown in Table 1, where we add "L" as suffix to the names of variants that use language embedding as attribute representation. Results show that with RBAs, users can obtain desired agent behavior much more efficiently than with PbRL. The upper row in Fig. 2 showcases how the user can obtain desired agent behaviors through sequences of attribute feedback with both methods. The interaction processes suggest that the attribute-parameterized reward is able to modify agent behaviors meaningfully in directions that are informed by RBAs. For full information of the interaction processes in all domains, we encourage the reader to check out the supplementary video.[1]

**Analysis of failure modes.** Results suggest that both methods have a lower success rate when trying to generate behaviors close to the two extremes (e.g., moving very softly or very recklessly). The bottom row in Fig. 2 shows two failure cases. In RBA-Global, the reward function fails to

---

[1]Supplementary video at https://guansuns.github.io/pages/rba

produce a behavior with a larger step size when we increase the target step size score in $v_t$ from 18.51 to 18.62. This disrupts the binary search process (Appendix A.3) and causes the system to get stuck. Similar failure patterns can also be observed in other domains. For example, in the Lane-Change domain, the agent fails to increase the distance to the following vehicle when the sharpness (proxy) score is 0.09. Since RBA-Global is composed of two learned models, namely the attribute strength estimator $\zeta_\sigma$ and the reward function $r_\theta$, we also examine them separately to see which module contributes more to the failures. By visualizing the outputs of $\zeta_\sigma$ and the corresponding proxy ground truth (Fig. 5 in Appendix A), we observe a positively correlated or sometimes linear relationship between them, suggesting that $\zeta_\sigma$ can accurately capture the attribute strength even at the two extremes. This observation verifies that the ineffectiveness of RBA-Global is mainly caused by $r_\theta$'s failure to recover behaviors specified in the target attribute-score vector $v_t$. Note that this is not surprising since $\zeta_\sigma$ only needs to learn one global ordering per attribute while $r_\theta$ has to learn almost an infinite number of orderings given various input targets $v_t$. Future work can explore better training paradigms and more expressive model architectures for $r_\theta$. In terms of the failure modes of RBA-Local, it sometimes fails to edit the behavior in the anchor trajectory $\tau_c$ and simply produces the same behavior as in $\tau_c$. As shown in the right bottom plot of Fig. 2, the agent gets stuck when the user wants to increase the softness when the agent is taking a large step (proxy score: 0.954). Also, RBA-Local tends to have a lower success rate than RBA-Global, indicating that its more complex formulation and architecture might affect its performance.

## 5.2 Additional Discussion

To get more insights into the number of labels needed to learn an accurate attribute ranking function or reward function in our methods, we conduct an extra experiment in which we train the model with different numbers of samples uniformly sampled from the training set, and evaluate each model on a held-out testing set. Results (Appendix A.5) show that to simultaneously learn two attributes, RBA-Global needs around 200 labelled trajectories (and the orderings among them) and RBA-Local needs around 200 $(\tau_c, \tau_t)$ pairs, which is a reasonable number. Also, the cost of learning a reward function is amortized when we continue to use it to support incoming users over its lifetime.

Recall that in the main experiment, we consider a trial as a successful one if the difference between the agent's behavior and the target behavior is lower than a threshold. One interesting thing to see is whether our methods can achieve even higher-precision control over the agent's behavior. Hence, we additionally experiment with a threshold value that roughly divides the attribute strengths into five to ten times more buckets. As expected, a more restrictive threshold reduces the performance of all algorithms, but our methods still have a significant advantage in terms of feedback efficiency. Results are shown in Table 2 in Appendix A. Note that this performance degradation was expected because the control precision we wanted to achieve in this experiment is higher than what we set in the training data (e.g., the precision corresponds to changes that are smaller than the minimally viable local changes defined in the training data for RBA-Local).

## 6 Conclusion

In this paper, we introduced the notion of relative behavioral attributes which allows users to provide symbolic feedback (i.e., their intent to increase/decrease attribute strength) to efficiently tweak and get desired agent behavior. We proposed two approaches and demonstrated their effectiveness through experiments in a varied set of domains. For future works, apart from the limitations we discussed earlier, it would be interesting if we could develop methods that combine the strengths of the two approaches proposed in this work. Also, currently, we use the sentence embedding of each attribute only as an alternative to the one-hot vector. However, it would be beneficial to make better use of the semantic structure inside the sentence embeddings.

Furthermore, it would be useful to explore the use of RBAs outside of continuous control tasks. For instance, for AI chatbots, we may construct rewards to capture not only binary attributes like helpfulness and harmfulness (Bai et al., 2022b) but also *abstraction levels* of the text (e.g., scientific concepts need to be explained differently to kids and researchers) or the *tonality* of the response (ranging from casual to a more formal or professional way). Inspired by recent attempts to build reward-driven vision models (Pinto et al., 2023), it would be also interesting to investigate whether RBAs can facilitate reward construction for fine-tuning models in computer vision tasks.

## ACKNOWLEDGMENTS

This research is supported in part by ONR grants N00014-16-1-2892, N00014-18-1-2442, N00014-18-1-2840, N00014-9-1-2119, AFOSR grant FA9550-18-1-0067, DARPA SAIL-ON grant W911NF19-2-0006 and a JP Morgan AI Faculty Research grant.

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

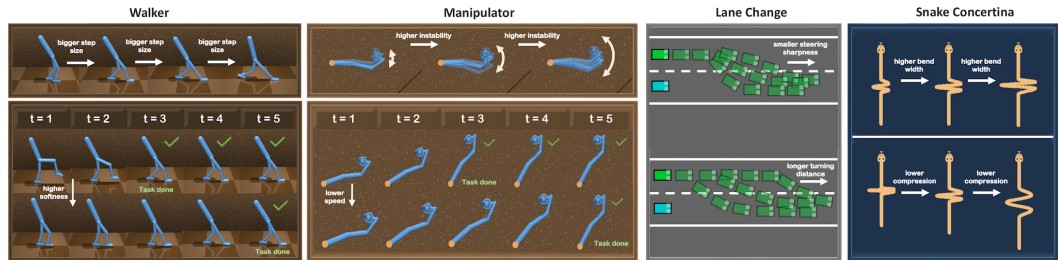

Figure 3: Visualizations of the evaluation domains and behavioral attributes.

# A APPENDIX

## A.1 DETAILS OF THE EVALUATION DOMAINS

**Walker.** Two attributes are considered in this domain: (a) step size; (b) softness of movement. To synthesize dataset and simulate human feedbacks, we use the moving speed and landing speed of the feet as proxies to measure the abstract concept softness or "sneaky". The environment is implemented based on the Walker-2d domain in the DeepMind control suite (Tunyasuvunakool et al., 2020).

**Manipulator.** We consider two attributes: (a) moving speed of the arm; (b) instability of the movement. The environment is implemented based on the Manipulator domain in the DeepMind control suite. We synthesize the behavior clips by hard coding the arm motion and by adding random noises in a controlled way. The purpose of conducting experiments in this domain is to demonstrate that our method can capture not only regular behavior patterns but also the irregularities. A Markov state is constructed by stacking five consecutive raw states of the environment to ensure it contains information about the irregularities.

**Lane Change.** Two attributes are used for evaluation: (a) the sharpness of steering: this attribute corresponds to how sharp a turn the agent makes while changing lanes; (b) distance to the following vehicle: this attribute is about the distance between our agent and the following car at the moment when our agent starts making the lane change. This environment is built on the highway environment in (Leurent, 2018). Note that the environment is an image based domain, so the objective here is to verify that our methods can be scaled to image inputs.

**Snake Concertina.** There are three relevant attributes in concertina locomotion: (a) width of the bend (i.e., the maximal width that the snake occupies); (b) compression (i.e., how much the snake's body is compressed when it is moving); (c) speed of movement.

## A.2 ALTERNATIVE WAYS TO USE THE ATTRIBUTE STRENGTH ESTIMATOR FUNCTION IN METHOD 1 (RBA-GLOBAL)

Recall that in RBA-Global, given a trained attribute strength estimator $\zeta_\sigma$ and a finite set of attributes $\mathcal{A}$, the agent behavior in any trajectory $\tau$ can be represented by an attribute vector $v(\tau) = \langle \zeta_\sigma(\tau, \alpha_1), ..., \zeta_\sigma(\tau, \alpha_k) \rangle$, where $k = |\mathcal{A}|$ is the number of attributes. If we assume the optimal policy is approximated via some parametric model (e.g., neural networks), we can actually skip the learning of the attribute parameterized reward function by viewing the attribute vectors as skill latent codes and learning a versatile policy conditioned on them. This is similar to the operations in (Wang et al., 2017; Peng et al., 2022) but with a more structured latent code. Also, one might create a sparse reward given at the end of episode by computing the distance between the extracted attribute vector and the target attribute vector. But such a reward is usually hard to optimize.

The main reason we choose to construct a reward function is that we find it more general, since rewards can be optimized not only by RL, but also by other optimization-based methods. Another consideration here is whether it is easier to learn a policy directly (e.g., via BC or IL) or to learn the reward first and then the policy (e.g., via IRL or PbRL). Prior empirical results suggest that the latter tends to be a more robust solution.

### A.3 FINDING THE TARGET ATTRIBUTE SCORES WITH BINARY SEARCH

Binary search is highly efficient because it narrows down the search space by cutting it in half at each step. In order to apply binary search in the attribute space, we need to maintain beliefs about the upper and lower bounds of each attribute. In our case, the upper and lower bounds can be initialized to the maximum and minimum attribute scores observed in the offline behavioral dataset $\mathcal{D}$, where the scores are given by $\zeta_\sigma$. At each query step, the agent presents to the human the behavior corresponding to the median attribute value, i.e., $\frac{\alpha_{upper} + \alpha_{lower}}{2}$, and updates the beliefs accordingly after getting the feedback. In the case of extrapolation, we can also go beyond the maximum and minimum attribute strengths, but this is no longer a binary search.

### A.4 ARCHITECTURES AND HYPERPARAMETERS

When one-hot representation is used to represent attributes, both $f_\sigma$ and $r_\theta$ in RBA-Global employ a 3-layer fully-connected network with 512 hidden neurons as the architecture. In RBA-Local, for the trajectory encoder, we use a 2-layer bi-directional LSTM with 128 as the hidden dim. The trajectory embedding, along with the input state and attribute, are fed into a 3-layer fully-connected network with 512 hidden neurons to compute the reward.

When sentence embeddings are used as attribute representation, for both RBA-Global and RBA-Local, we increase the number of hidden neurons in fully-connected layers from 512 to 1024 due to the increase in the size of attribute embedding (size 768).

### A.5 NUMBER OF TRAINING SAMPLES NEEDED IN OUR METHODS

Fig. 7 and Fig. 8 show the performance (on a held-out testing set) of Method 1 (RBA-Global) and Method 2 (RBA-Local) when different numbers of training samples are used. For RBA-Global, given an ordered trajectory pair $(\tau_1 \succ_\alpha \tau_2)$, the ranking function is converted into a binary classifier that predicts the ordering of the given pair. The performance is measured in terms of the accuracy of this binary classifier. Similarly for RBA-Local, the performance is measured by converting Equation 5 to a binary classification problem where the function predicts whether a trajectory is the target trajectory or not. Note that the sample complexity of the two methods is not directly comparable because their training samples are in different formats.

### A.6 DISCUSSION ON UNSUPERVISED DISCOVERY OF BEHAVIORAL ATTRIBUTES

As a preliminary attempt in this study, we explored the possibility of unsupervised discovery of concepts or properties. We applied a state-of-the-art disentangled sequential variational autoencoder method, C-DSVAE (Bai et al., 2021), to learn to encode behaviors in the offline behavior dataset $\mathcal{D}$. Though the behavior/motion embeddings given by C-DSVAE are able to cluster visually similar traces together, it still has difficulty capturing subtle but meaningful differences in behaviors. More importantly, it fails to establish any meaningful ordering among behaviors. As an example (Fig. 6), we visualize the variations encoded in a specific dimension in the latent space of the Lane-Change domain. This observation is consistent with the current trend in vision-text research, which suggests unless additional supervision signals are provided, representations developed by neural networks are not guaranteed to capture semantics that make sense to humans. This preliminary study confirms the necessity to employ supervised learning for behavioral attribute modeling.

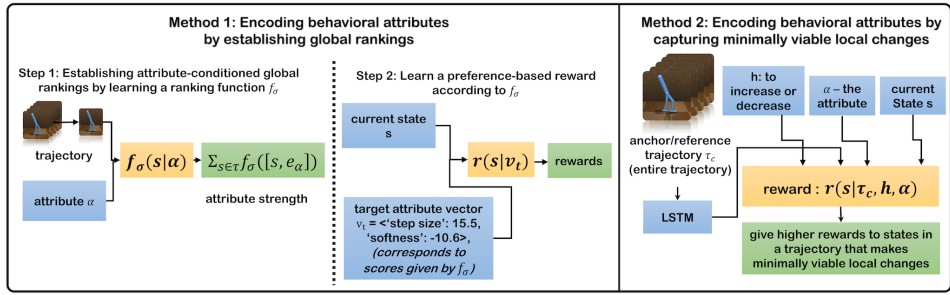

Figure 4: Overview of Method 1 (RBA-Global) and Method 2 (RBA-Local).

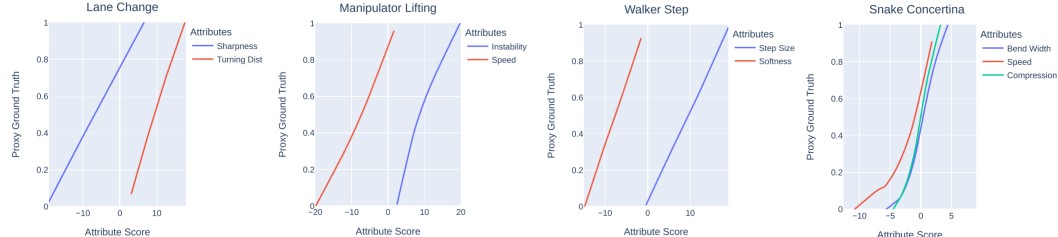

Figure 5: The relationship between $\zeta_\sigma$ and the hand-designed proxy ground truth. The attribute strength scores are computed with one-hot vectors used as the attribute representation. We can observe a similar pattern when language embeddings are used.

| Method | Lane-Change | | Manipulator | | Snake | | Walker | |
|---|---|---|---|---|---|---|---|---|
| | SR | AF (std) | SR | AF (std) | SR | AF (std) | SR | AF (std) |
| RBA-Global | 0.60 | 6.75 *(2.98)* | **0.80** | **7.00** *(2.45)* | **0.55** | **8.00** *(3.67)* | 0.70 | 7.29 *(2.63)* |
| RBA-Global-L | **0.7** | **6.79** *(1.66)* | 0.75 | 5.47 *(3.2)* | 0.25 | 10.4 *(6.02)* | **0.75** | **6.00** *(3.59)* |
| RBA-Local | 0.25 | 5.00 *(0.63)* | 0.45 | 19.44 *(5.57)* | 0.30 | 8.16 *(3.18)* | 0.40 | 9.37 *(4.15)* |
| RBA-Local-L | 0.4 | 8.63 *(2.45)* | 0.30 | 10.5 *(5.5)* | 0.50 | 7.2 *(4.621)* | 0.40 | 6.25 *(2.48)* |
| PbRL | 0.35 | 286.22 *(167.49)* | 0.15 | 81.33 *(66.97)* | 0.05 | N/A | 0.35 | 288.0 *(143.45)* |

Table 2: Results on controlling the agent's behavior with higher precision. SR - Success Rate; AF - Average Feedback (when success); L - Language

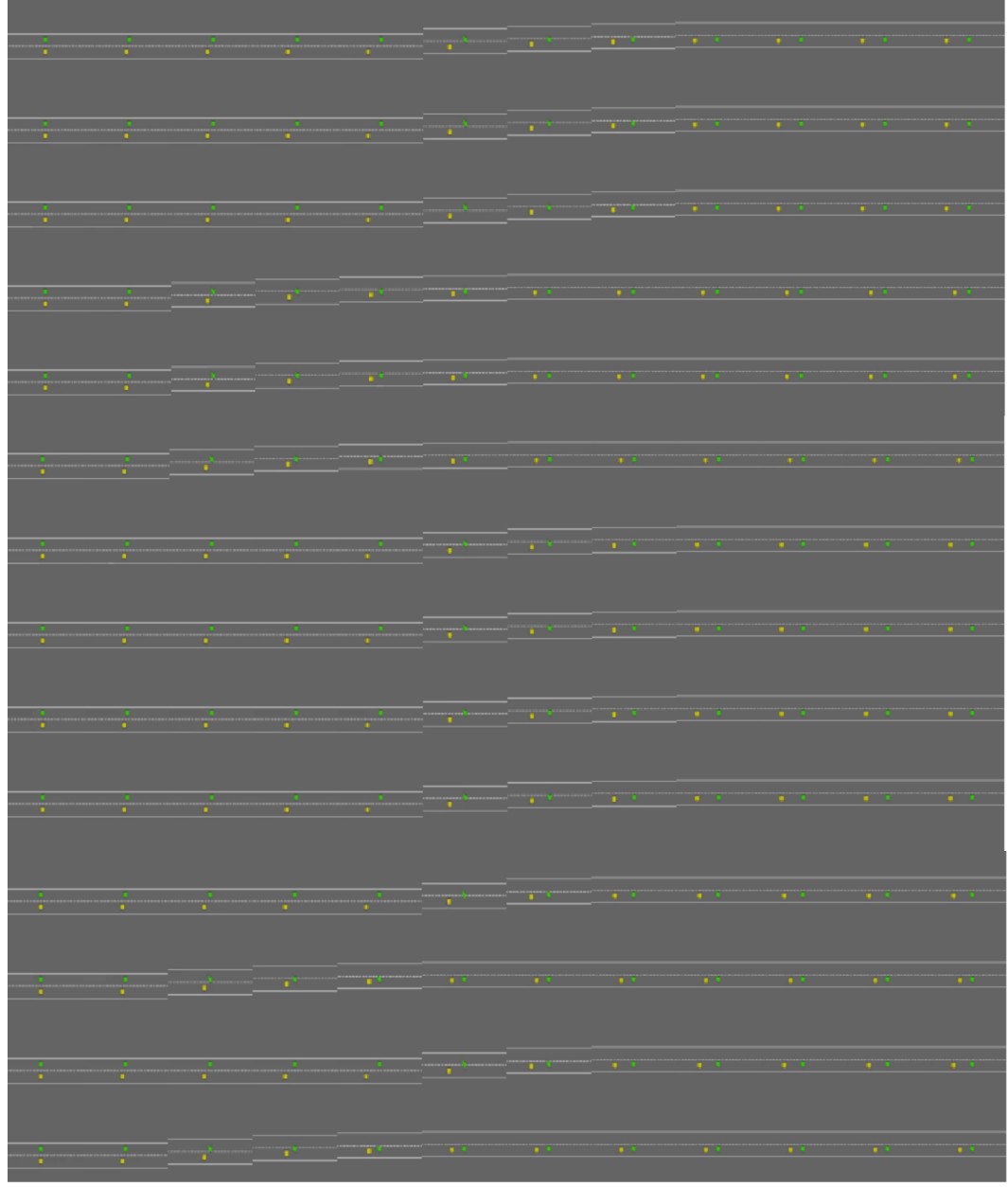

Figure 6: A failure case of unsupervised concept discovery. Each row corresponds to a behavior trace.

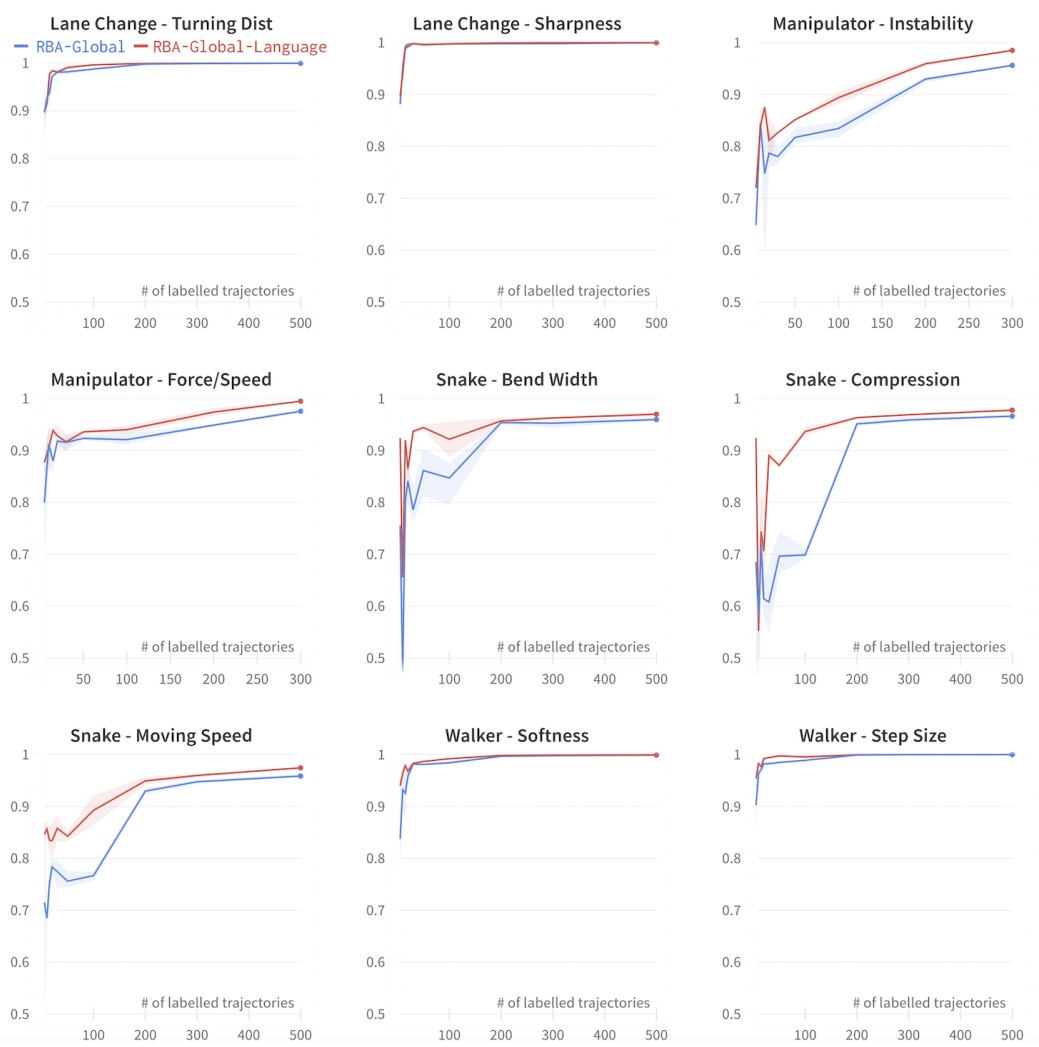

Figure 7: Performance of the ranking function in Method 1 (RBA-Global) versus # of training samples.

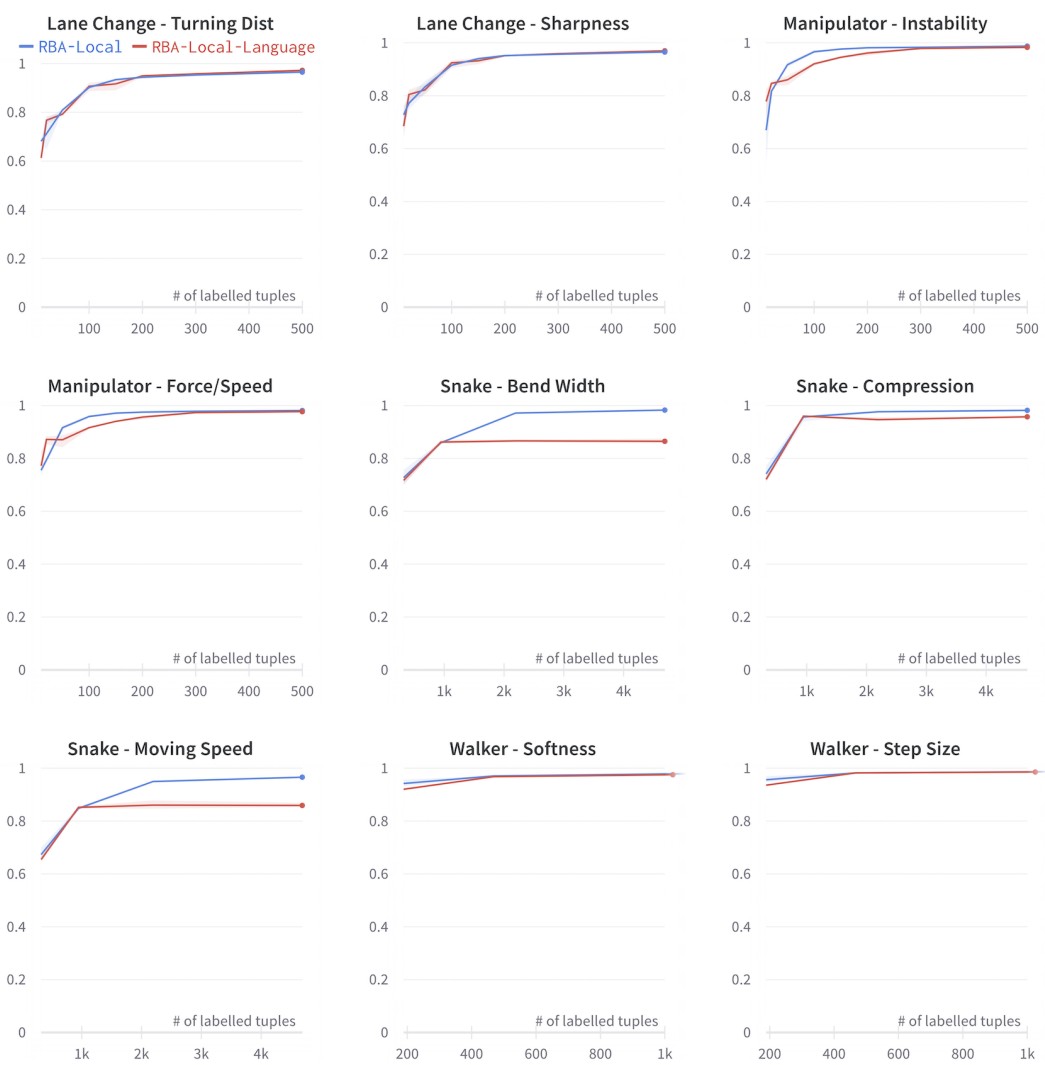

Figure 8: Performance of Method 2 (RBA-Local) versus # of training samples in Method 2 (RBA-Local).

