# OpenReview forum: "Relative Behavioral Attributes: Filling the Gap between Symbolic Goal Specification and Reward Learning from Human Preferences"
_ICLR.cc/2023/Conference — ICLR 2023 poster_

### Official Review · Reviewer_mgGn · 2022-10-22

**Confidence:** 4
**Correctness:** 3
**Technical Novelty And Significance:** 3
**Empirical Novelty And Significance:** 3
**Recommendation:** 6

**Clarity, Quality, Novelty And Reproducibility:**

As said in the above, the clarity of the approach is far from optimal, and a good rewriting of the core ideas of the paper should be done. The quality of the work is ok; I think that technically the idea is sound, but it needs a more solid evaluation (and possibly comparison to more work). Novelty is ok as well; the core idea of learning reward functions from preferences over traces is not new, but this particular way of reranking trajectories is, and it seems to be medium significant although it is not entirely clear since the related work is not done optimally. I think that novelty in terms of the title, as in "providing a new middle ground to reward functions" is a bit pretentious given what is in the paper as description of this sub-field. The reproducibility of the paper should be fine, based on this description, and if code would be available.

**Strength And Weaknesses:**

Reward functions are the core of RL and work in finding new/different ways to incorporate and specify them is interesting. The strength of the paper is that it is a quite practical idea for a particular kind of reward function learning (and specification) in RL. The approach is intuitive, and amenable to all kinds of extensions. I think that the outcomes of the experiments show that the approach works.

The paper has, as a research paper, a couple of weaknesses that are not all easy to address.
- An overall weakness is the text of the paper itself, including the formalizations. As far as I understand the paper, the work builds on typical (and recent) approaches leveraging Bradley-Terry-style models like on page 4 that can turn datasets of traces with preferences over these traces into a reward function that can be utilized to tune behavior towards those preferences. The main idea of the paper is to place either a vector of relative attributes in that ranking, or separate attributes (in the local model) and to derive a new reward function that incorporates the intended values (or direction of values) of those attributes. In both the local and the global model, the user can adjust the value of attributes such that the behavior of the agent changes in some intended direction. This is a fairly simple story and setup, but I had a hard time reading through all the verbose texts (already starting in Section 1) to find out what was happening. I'd advise a full rewriting and restructuring of the paper to reflect this. Also, the formalizations can then be made more precise and complete, and all this would help to create a much better description of the method. Also define simple things like "skill" or "skill behaviors", which seem to be assumed to be known, but which have very specific meanings in sub-areas of RL. Also, formalize the main aspects of the "re-weighting"/"re-ranking" of the traces better. The precise setting in sections 4.2. and 4.3. stays too implicit formally. Things like "attribute encoder" are used without explanation or formalization. Maybe give a couple of good examples in the beginning to sketch te idea more precisely. Also, the paper is not polished: several small errors (like there is no Table 5 on page 9, "sample at large" on the last line of page 8, informal style for "don't" instead of "do not", double words "the the" on page 5, and ungrammatical sentences like the first sentence of the 2nd par of Section 4.2. around "than", etc). Overall, I think the core of the paper is not described well, and this hinders a good evaluation of the technical parts of the paper.
- The positioning of the work is done poorly. First of all, already in Section 1 we see many different aspects, and levels of reward function definitions, but nothing is really explained nor compared. I think, since nothing is really explicitated around this "lingua franca" (page 1), it would be much better to say something like "sometimes one has some ideas about aspects of a full behavior that should be changed, a property of the behavior, and we would like to increase or decrease it, and for this we introduce relative attributes, a little in direction of how people do style transfer in deep learning". The title promises much more than is in the paper now. There is nothing really about symbolic goal specifications as in (temporal) logic, where relative attributes as presented in this paper would be something like a maintenance goal. Furthermore, relative attributes are limited (as in the experiments) to a general number about the trace as a whole, which should be discussed. The experiments too all use these types of attributes. All these aspects should be related to the literature, which is not really happening in Section 2: we see some general references and some more specific, but it is hard to what is the state-of-the-art and what is the precies positioning of this paper, simply also because the section before it does not really define the setting well. The related work section should be much improved, and include things like reward functions (symbolic, preference rankings, but also inverse RL which is very related) but also things like style transfer in RL and other related topics. Also, some of the paper sounds too pretentious, like "lingua franca" and "extremely efficient manner" and "superior performance", etc.
- The experiments are showing viability but not much more. From the technical description I get the feeling that this approach will work in some way (since, more information goes into the ranking and that seems to be a sound idea) but not how well, nor how it "really" works. The only baseline used is PbRL but this is not explained at all. The comparison is based on a couple of assumptions on how to use both (PbRL and the new technique) but this is not motivated enough. An important choice are the amounts of feedbacks (30 and 1000) but it is unclear how they influence the results, which are based on "proximity" to some good solution, and this is the only thing we get to see in Table 1. The number of domains considered is good, but the set of experiments is very limited and the analysis very short (one paragraph). I think the experimental evaluation should be greatly extended (not necessarily in the amount of domains or experiments, but in terms of analysis and sensitivity analysis) which is possible if the first half of the paper is written more concisely. I like the general embedding approach where also language descriptions can be used, but also here I'd like to see much more information and analysis (and examples). Basically, I'd like to get more detailed (visual) analysis of "how" behaviors are tuned based on the attributes in a specific domain (how fast, how well, artifacts, visibility of changes, correspondence between attributes and measured proxies, and so on).

**Summary Of The Paper:**

This paper takes an existing reward learning paradigm in reinforcement learning (RL) that can learn reward functions from preferences over traces (of states) and refines it by allowing to target specific "relative attributes" (properties of a trace usually) to be optimized. The incorporation of the relative attributes enables to tune some "general" properties (usually) of traces, for example the smoothness or speed of certain movements of a robotic manipulator. The approach comes with two different ways to incorporate (and tune by a user) the relative attributes: one global technique that focuses on a vector of values, and one local technique where one attribute can be optimized. The approach is tested on a couple of domains and various relative attributes.

**Summary Of The Review:**

Nice practical idea in the sub-area of RL to learn from user preferences. The description (both text and formalizations, related work) needs quite some work, as well as the experimental analysis which should be extended in terms of analysis and description.

---

> ### Author Response · Authors · 2022-11-15
> **Response to Review**
>
> We thank reviewer mgGn for the thoughtful feedback. We are happy that the reviewer found our approach to be intuitive, practical, and amenable to various extensions. We have incorporated the reviewer’s constructive feedback to the best of our ability into the revised version of the paper. In the following, we will address specific comments and explain the corresponding revisions & clarifications.
>
> ### **1. Improvements to the presentation of the methodology**
>
>
> > **1.1. A summary of the revisions**
>
> We made a series of revisions to the manuscript according to reviewer mgGn’s feedback. We kindly refer reviewer mgGn to Point 1 in the common response for a detailed list of the added analysis.
>
> We also encourage reviewer mgGn to check out the supplementary video, in which we presented both the success cases and failure cases of our methods. **The video can be found in the supplementary material or through the link: https://easyupload.io/cobar0 .**
>
> We would like to thank the reviewer in advance for spending the time to read the revised manuscript and watch the supplementary video.
>
>
> > **1.2. Define simple things like "skill" or "skill behaviors", which seem to be assumed to be known, but which have very specific meanings in sub-areas of RL**
>
> The exact definition of skill was presented in Sec. 3 in our previous manuscript.  We are aware that under different problem settings, “skill” has been abstracted either as “options” in a Semi-MDP or as a task in a standard MDP. Here, we adopt the standard MDP formalization as described in Sec. 3.
>
>
> > **1.3. Several small typos**
>
> We apologize for those typos/minor inconsistencies;  we have now fixed them.
>
>
> > **1.4. Some of the paper sounds too pretentious, like "lingua franca" and "extremely efficient manner" and "superior performance", etc.**
>
> We corrected the mentioned expressions:
> - “Lingua franca” -> “shared vocabulary between humans and inscrutable models”
> - “Extremely efficient manner” -> “more efficiently than PbRL”
> - “Superior performance” -> we removed this
>
>
> ### **2. Extend the analysis in the Experiment section**
>
> > **2.1. The experiments are showing viability but not much more**
>
> We have made a series of changes in our revised paper to improve conciseness. This allows us to include a more comprehensive analysis of the proposed framework in the experiment section. We kindly refer reviewer mgGn to Point 2 in the common response for a detailed list of the added analysis.
>
>
> > **2.2. PbRL is not explained at all**
>
> In Section 3 of the previous manuscript, we described PbRL in detail. To make it more noticeable, in the revised version, we explicitly mention that “the latter is often referred to as preference-based reinforcement learning, or PbRL for short.”
>
>
> > **2.3. An important choice is the amount of feedback (30 and 1000) but it is unclear how they influence the results**
>
> Ideally, a trial should terminate within a user-affordable number of feedbacks. We can imagine that the end users will lose patience quickly after providing a small amount of feedback (like 50 times). But purely for evaluation purposes, we set the user-affordable number to an upper bound of 1000 (in fact, 1000 is also an impractical number for any real-world use case. If the reviewers see the need to better reflect reality, we can further set the upper bound to a smaller number and provide the updated results).
>
> Regarding 30, it was based on the observation that when our system succeeds, it usually finishes the job within 10-20 feedbacks; and when it fails, it usually gets stuck at some behavior and fails to further change the behavior (we weren’t able to mention this due space limit previously, but we have addressed this in the revised version). Hence, 30 was used as a way to do early stopping and to save time. To avoid confusion, in the revised version, we simply state 1000 is used for all the methods.

---

> > ### Author Response · Authors · 2022-11-15
> > **Response to Review (continued)**
> >
> > ###  **3. The positioning of this work can be better explained**
> >
> > > **3.1. First of all, already in Section 1 we see many different aspects, and levels of reward function definitions, but nothing is really explained nor compared**
> >
> > We received divergent feedback on the Intro section and the positioning of the work. We agree that our previous Introduction section (Sec. 1) tends to assume methods like PbRL and symbolic reward specification are already known to the readers, thus omitting many details and possibly causing confusion. We hope that the revised manuscript and the added Humanoid example do a better job in describing the background.
> >
> > Apart from the revision, we also wanted to provide some clarifications here. **The most important motivation for this work is the slow progress/advance in PbRL**. In many previous PbRL works, we have seen how pairwise trajectory comparisons together with a parametric reward function allow non-expert users to convey complex objectives that “conventional” symbolic goal specification methods cannot do. However, the high data complexity of PbRL has always been a bottleneck to its applicability, and **we believe that the PbRL community has been too resistant to accepting any symbolic input from humans**. What we are trying to do in this paper is to show that, by allowing humans to indicate their preferences in terms of symbolic concepts (i.e., taking a step back towards the more “traditional” symbolic reward specification), we can easily deliver an efficient and practical solution to construct parametric rewards and enable non-expert users to customize the agent behavior.
> >
> > We would also like to note that there is no strict “state-of-the-art” in reward specification. The choice of method for goal specification depends on the nature of the underlying task and the user’s preferences. For example, in the newly added Humanoid example (Sec. 1), if the user only wants the robot to barely move forward, it might be easier for the user to construct a symbolic reward. On the other hand, if the user’s objective is too complex and there isn't any well-established concept that can describe the user’s requirements (e.g., some novel motion style), the user may find it easier to spend time providing preference labels over queried trajectories. Our methods are targeted at scenarios wherein the user’s objective is complex (so a full symbolic specification is not applicable here) but still describable in terms of nameable concepts. We believe this setting is quite practical, and it should cover a large body of real-world use cases.
> >
> > Moreover, since there is no strict “state-of-the-art”, we encourage the reviewer to focus more on the performance of our proposed framework itself. It is not surprising that our approach is able to outperform PbRL as we are accepting richer inputs from the users. What we want to showcase through the experiments is that, by sacrificing a little bit of generality compared to PbRL (e.g., by assuming the user’s objective is partially describable and assuming the attributes of interest are captured in a parametric reward), we are able to deliver a practical method for goal specification which only needs a reasonably small number of feedbacks from the end user.
> >
> >
> > > **3.2. There is nothing really about symbolic goal specifications as in (temporal) logic**
> >
> > We would like to clarify that, the “symbolic” here is **describing how the user is interacting with the system in specifying the reward**. This is not to be confused with “reward computation in a symbolic way” (e.g., in reward specification via temporal logic, the exact reward value is determined by some symbolic expression). In this work, we emphasize more on how to allow the users to express their preferences symbolically (i.e., in terms of concepts). This is parallel to how the reward is computed. In fact, by interacting with the user symbolically and computing the reward in a parametric way, we essentially bring the best of the more “conventional” symbolic specification (e.g., through temporal logic) and PbRL.
> >
> >
> > > **3.3. Relative attributes are limited (as in the experiments) to a general number about the trace as a whole, which should be discussed**
> >
> > We have added a discussion about how we formulate a relative attribute as a mapping between a trajectory to a real-valued number. But we don’t see this as a limitation, because to learn/approximate such a mapping, we only require the agent builder to provide the orderings among trajectories, which is a general enough setup. This formulation is sufficient to capture all possible attributes if the task is modeled as an MDP and the agent is learning a policy for the MDP.

---

> > > ### Author Response · Authors · 2022-11-15
> > > **Response to Review (continued)**
> > >
> > > > **3.4. The related work section should be much improved, and include things like inverse RL and style transfer in RL and other related topics.**
> > >
> > > We have added a discussion about IRL in the revised related work. We believe what reviewer mgGn meant to say with “style transfer in RL” is essentially the works that learn diverse skills or motion styles, which are already covered in our manuscript.
> > >
> > > **Again, we thank the reviewer in advance for spending the time to read this long response.**

---

> > > > ### Comment · Reviewer_mgGn · 2022-11-17
> > > > **response 2**
> > > >
> > > > Thanks for all the long clarifications, and the answers. I have taken a look at the revised paper so far, and the general remarks. I still think the framing (and thus the related work) is not optimal. I could imagine starting with a title "Relative Behavioral Attributes: more finegrained control in preference-based reinforcement learning" or something along those lines. I think that the whole "symbolic" aspect (and related references to things dealing with temporal logic) can be discarded which would make the story easier, because you are basically extending PbRL with something that gives you more options (with some meaningful naming of these options) to finetune behaviors. I read the original paper on physical paper, and I wrote as a summary on the first page "Framing of what it is must be done much better". I think you have already done some good steps, but also in the answer above a lot of things and arguments are mixed which make the story more complicated than needed. I think that the framing could also be helped by saying more about style transfer (like) things to describe the intuition of RBAs.
> > > > But, the current revisions to the text, the analysis, the explanations of methods, is for me enough to raise my score from a weak reject to a weak accept. I still see remaining issues with related work, analysis and writing, but for the revised form (and I really hope more revisions can and will be done to improve even more) I am leaning to accept.
> > > > Of course, I agree there is no strict state-of-the-art, but I do strongly believe that by framing your method --exactly-- where it needs to be in the scientific landscape, one can more easily and more quickly see what it is, what the contributions are, and how the local landscape has evolved because of the results in the paper. Examples of where the original paper could be improved were 1) if the framing in the introduction is done more concisely, the related work can exclude symbolic value functions for the most part for example, and 2) the inclusion of IRL etc (which is now done), 3) if the connection to the right formalizations (MDPs, options, still should be done in my opinion) and the right results for PbRL, then the analysis (which is updated now) can give more insights into how this new method changes the state-of-the-art.

---

> > ### Comment · Reviewer_mgGn · 2022-11-17
> > **small response 1**
> >
> > Thanks for all the detailed answers and clarifications. Just two things to mention here based on this:
> > 1) Good to explicitly connect PbRL to Section 3. I hope you see the confusion in the original paper, since Section 5.1. mentioned "THE PbRL algorithm proposed in..." whereas Section 3 mentioned aspects of "such algorithms" (and some of the same references) without mentioning PbRL.
> > 2) The answer about "skills" is equally imprecise as the original text. The original text defines a skill as "a solution that solves an episodic task in an.... MDP..." (which in itself is already vague and verbose). Then there is a "policy", but the "skill" is performed, and so on. This is confusing, and why not doing it just like options? So... take a standard MDP, and learn subpolicies (skills, options) as specific tasks in an MDP, with start/finish conditions and so on? Using "skill behaviors" later on, and other terms, does not help either. My message here is that simplicity is key, and can help in carving out the main contributions of the paper a lot.

---

### Official Review · Reviewer_Ry2j · 2022-10-24

**Confidence:** 3
**Correctness:** 4
**Technical Novelty And Significance:** 3
**Empirical Novelty And Significance:** 3
**Recommendation:** 6

**Clarity, Quality, Novelty And Reproducibility:**

The paper is very well written and easy to follow. The motivation is incredibly clear and the introduction does an excellent job of characterizing the existing state of the art, how it is limited, and proposing an alternative approach that overcomes the data inefficiency of competitive approaches.

Experiments are fairly comprehensive within the context of the provided benchmarks (a reasonable set that demonstrate the versatility of the approach). The approach is novel, expanding on a problem studied in the literature, but proposes a new approach (loosely inspired by recent work in recommender systems) to support preference-aligned behaviors with few demonstrations.

There is no reproducibility statement and no code is provided.


**Strength And Weaknesses:**

Overall, the paper is fairly strong. The authors introduce two different approaches to using their approach and do a good job discussing the relative advantages of each and provide suitable performance analysis. Experiments are well-suited to understand the problem and cover multiple domains of interest to this community. PbRL is also a solid baseline, and the comparison to PbRL does a good job at highlighting why an alternative approach is necessary.

One part of the paper that could benefit from further discussion is where the approach is limited or is unlikely to succeed. For example: what if the attributes are limited or poorly chosen? How would performance change (presumably decrease) if an attribute is omitted? Further experiments are likely unnecessary, but adding such details (at least some of which should appear in the main body of the paper, even if others appear in the appendix) would help clarify when the proposed approach(es) are or are not well-suited to the proposed work.

Could the authors comment in more depth on what a typical failure mode looks like? There is some general discussion of why failure modes occur: namely what seems to be the limited gradient of the ranking function near the extreme values. However, some additional figures would improve understanding. Is the failure that the corrections do not change behavior? Some additional (qualitative) visuals or analysis would help clarity.

Finally, though it is straightforward to understand how embedded language descriptions could be used in place of a one-hot vector representation for the attribute vectors. However, it is not clear to me why the language-input models (in most of the experiments in Table 1) is competitive with or even frequently improves upon the performance of the non-language counterparts, since the non-language versions are trained using a representation specifically chosen by the designers (in this case, the authors) for this purpose. Could the authors clarify what the source of this improvement might be?

Other smaller comments:
- The abstract is quite long, and it might be good to remove some of the high-level context in the interest of conciseness.
- The Walker column in Table 2 seems to be incorrectly highlighted: RA-Global-L seems to perform best across all metrics.


**Summary Of The Paper:**

This paper proposes "relative behavioral attributes" an approach to allow non-expert end users to quickly (i.e., with limited data or demonstrations) correct robot/autonomous behaviors according to their preferences. The central approach involves learning (and later retraining) a ranking function that yields how strongly a particular attribute appears in a trajectory or behavior. Designers provide these attributes and labeled data for a ranking of how strongly attributes appear in each trajectory. During deployment, an end user can specify which behaviors they prefer through limited interactions---involving stating that a trajectory is preferred over another---and this data is used to train a preference-based reward model that uses the learned attribute models. Because trajectories are characterized in terms of a small set of attributes, retraining happens quickly, and far fewer feedback examples are required to achieve the desired behaviors. The authors show good performance on four different domains and significantly faster behavior change compared to the recent PbRL approach.


**Summary Of The Review:**

The paper is clear and delivers on most of its promises. I have a few questions and comments that the authors should address so that the paper can be maximally useful for the community, though they are relatively small.

---

> ### Author Response · Authors · 2022-11-15
> **Response to Review**
>
> We thank reviewer Ry2j for the valuable comments. We are happy that reviewer Ry2j found our work to be novel and the paper to be well-written. Here are some clarifications on some of the comments and raised questions:
>
> > **1. In-depth analysis of the failure modes**
>
> We have made a series of changes in our revised paper to improve conciseness. This allows us to include a more comprehensive analysis of the proposed framework in the experiment section. We kindly refer reviewer Ry2j to Point 2 in the common response for a detailed list of the added analysis.
>
> We also encourage reviewer Ry2j to check out the supplementary video, in which we presented both the success cases and failure cases of our methods. **The video can be found in the supplementary material or through the link: https://easyupload.io/cobar0 .**
>
> We would like to thank the reviewer in advance for spending the time to read the revised manuscript and watch the supplementary video.
>
>
> > **2. How would the performance change if an attribute is omitted or the set of attributes is limited?**
>
> Please note that the goal of collecting feedback from humans is to align the agent’s behavior to the user’s preferences. Hence, if an attribute is omitted by the user, it only means that the attribute is irrelevant and the user is okay with any strength of the attribute’s presence in the agent’s behavior. This thus won’t affect the “performance” of the system. And we would imagine that in this case, the system will even need less feedback from the human because there are fewer targets to reach (i.e.,# of attributes to tune) for the agent.
>
> Regarding the second condition (i.e., if the user finds that the set of attributes is limited), this corresponds to the case that the user wants to extend the set of attributes. As discussed in Sec. 4.1, the user will need to label additional data such that the system can learn to model the novel attribute (basically doing the same job that the AI agent builder does). Note that the cost of providing training labels for this novel attribute will be amortized over time if other future users also want to customize the new attribute.
>
>
> > **3. The benefit of using sentence embeddings as attribute representation**
>
> We kindly refer reviewer Ry2j to Point 3 in the common response for a detailed list of the added analysis.

---

> ### Author Response · Authors · 2022-11-22
> **Follow Up**
>
> Dear Reviewer Ry2j,
>
> We are just checking in to see if you had a chance to look at our revisions and responses, and if you had any further questions. Thanks once again for your support of this work.
>
> Best regards,
> Paper4982 Authors

---

### Official Review · Reviewer_5ZSD · 2022-11-03

**Confidence:** 2
**Correctness:** 4
**Technical Novelty And Significance:** 4
**Empirical Novelty And Significance:** 4
**Recommendation:** 8

**Clarity, Quality, Novelty And Reproducibility:**

I think everything seems OK, but I don't work in this area myself, so I'm not sure what kinds of details I'd want to know if I was going to try to literally reproduce the experiment myself.

**Strength And Weaknesses:**

Strengths
-----------
- Novelty : there is surprisingly little published academic work on how to design reward functions. As the manager of a team using RL for a real product, I was surprised to find that "reward function" is not in the index of the Sutton and Barto book, and not a page in Wikipedia. So this paper contributes to a very underdeveloped area.
- Usefulness: my experience working in industry is that the biggest barrier to using RL for real products is not the performance of the RL algorithm itself, it's challenges outside of core RL, like sim to real transfer and how to design the reward function and evaluate performance of the agent. Real-world products interacting with human users have much more complicated reward functions than 0/1 winning or losing Go. So this paper contributes to an area that is important for practitioners in industry.

Weaknesses
--------------
- Experimental results: the experiments here are all on synthetic data and toy tasks. I'm not super optimistic that the methods proposed here would help my RL team on a real product. However, I think this is acceptable due to the limited amount of literature on the topic / I think other papers accepted in this area share this weakness.



**Summary Of The Paper:**

The paper introduces a new method for training RL agents with human feedback. In prior work, agents are trained either with a fully programmed reward function (which is hard to program correctly) or are trained with low bandwidth human feedback (e.g. binary feedback on which of two trajectories was better). This work proposes learning a set of attributes that describe trajectories, then getting human feedback on how a trajectory should change, in terms which which attributes to increase / decrease. This gives a higher bandwidth method of human feedback, leading to fewer rounds of feedback before successfully learning a concept.

**Summary Of The Review:**

I think this paper provides a reasonable step forward in an important subfield that is highly underdeveloped.

Statement on confidence: I haven't worked on this topic or even topics all that closely related to this firsthand, so it's reasonably likely that I've missed something important, particularly in terms of related work.

---

> ### Author Response · Authors · 2022-11-15
> **Response to Review**
>
> We thank reviewer 5ZSD for the valuable comments. We are happy that reviewer 5ZSD found our work to be novel. Here are some clarifications on some of the comments and raised questions:
>
> - The evaluation domains used in this paper are from some of the standard benchmarks widely used in academic research (e.g., DeepMind control suite and the highway driving environment). We understand the gap between industrial and academic environments. In the future, we would definitely like to demonstrate our methods in more complex real-world domains.
>
> - We also encourage reviewer 5ZSD to check out the supplementary video, in which we presented both the success cases and failure cases of our methods. **The video can be found in the supplementary material or through the link: https://easyupload.io/cobar0 .**
>
> - The goal of this work is indeed to come up with an approach that is practical and can be applied to real-world products supporting real users. We believe the proposed relative attributes approach can be a useful tool due to its low feedback complexity. We hope that you can try out our proposed methods in the products that you are building someday.

---

> ### Author Response · Authors · 2022-11-22
> **Follow Up**
>
> Dear Reviewer 5ZSD,
>
> We are just checking in to see if you had a chance to look at our revisions and responses, and if you had any further questions. Thanks once again for your support of this work.
>
> Best regards,
> Paper4982 Authors

---

> > ### Comment · Reviewer_5ZSD · 2022-11-23
> > **Yes, I've read the response**
> >
> > Confirming that I've read the authors' response. I still think 8 is an appropriate score, the author's response is consistent with what I already understood. The other reviewers have lower scores---I've only skimmed their criticisms, and might reach a sharper opinion if the ACs asks us to reach a consensus---and based on the limited effort I've put into reading the other reviewers' comments I think their 6s and my 8 represent legitimate spectrum of opinion / I think an average score of ~6.7 is probably fair.

---

### Author Response · Authors · 2022-11-15
**Common Response to All Reviewers**

We thank the reviewers for their thoughtful comments. This common response serves as a summary of the revisions we made to the paper during the rebuttal. We also provide clarifications on some common comments here. We would like to thank the reviewers in advance for spending the time to read our response and the revised manuscript.

**1. Improvements to the presentation of the methodology**

We note that reviewers 5ZSD and Ry2j were largely satisfied with the writing of our paper and found the introduction section to be very effective in motivating our work. Nevertheless, we paid serious attention to reviewer mgGN’s comments and made several revisions to further improve the writing. Here is a detailed summary of the revisions we made (the important changes are highlighted in blue text in the revised paper):

- We shorten the abstract to make it more succinct.

- In Section 4.2 (Method 1) & Sec. 4.3 (Method 2), we trim some less important explanatory texts and use formalism as much as possible to present the methods more concisely. As a result, Sec. 4.2 and 4.3 are shorter now. We hope these changes will improve the readability for the readers.

- Restructuring Sec. 4.2 (Method 1): Instead of introducing each component sequentially, in the updated version, we first give an overview of all the components, such as the inputs to each module, and the functionality of each component. We also use the humanoid household robot as a running example to illustrate how each component is supposed to be used. We only start discussing the details of each component (e.g., formats of the training samples, and detailed architectures) after this leading overview paragraph.

- Improving Sec 1. Introduction: We revise the second paragraph to include **a detailed example** (the Humanoid control task from the DeepMind control suite) to illustrate what rewards can and can’t be specified in a closed-form manner. We give an example of a possible symbolic reward, so that the reader can get a sense of what symbolic reward specification means and how it is usually used. Then we discuss why reward specification via pairwise trajectory comparisons, together with a parametric reward, can be a more general method for any task. But treating every task at hand as a tacit-knowledge one and limiting reward specification to binary comparisons can be unnecessarily inefficient. A large body of tasks in real life involve a mixture of tacit and explicit knowledge. Hence, to improve efficiency and user experience, we argue that whenever the users are able to express their preferences in terms of nameable concepts, they should be allowed to do so. This motivates the development of relative attributes.

- We avoid the use of the term “attribute encoder” (which is very vague).  We now use “attribute-parameterized reward function” and say that the reward “can encode complex task knowledge while allowing end users to freely increase or decrease the strength of certain behavioral attributes.“




**2. In-depth analysis of the failure modes**

We agree that there should have been some in-depth analysis of the experiment results. We weren’t able to do that in the previous version due to limited space. In the revised version, we reduce the length of previous sections (i.e., describing the proposed methods in a more concise way as suggested by reviewer mgGn) to have more space for in-depth discussions.

We add the following points in the revised paper to further improve the analysis of our experiments:

- An annotated video that contains the success and failure cases of both methods in all the domains. This video shows how the agent behaviors change according to user feedback and displays the corresponding attribute strength given by some hard-coded proxy measure. We also present the detailed information of the agent such as the beliefs of target attribute strengths (i.e., upper bound and lower bound) that are maintained by the Method 1 agent. **This video can be found in the supplementary materials or through the anonymous link: https://easyupload.io/cobar0 .** Please note that this anonymous link is only valid for 4 weeks. When this link expires, please feel free to request a new one from us.

- A visualization of the interaction process with proxy measures. The visualization can be found in the new Fig. 2.

- A discussion on the failure modes of both Method 1 and Method 2. We also provide examples of failure cases. We believe this would give the readers a much better sense of the limitations of our methods.

- Further, to examine which modules contribute more to the failure cases in Method 1, we visualized the outputs of the attribute strength estimator (Fig. 4 in Appendix).

---

> ### Author Response · Authors · 2022-11-15
> **Common Response to All Reviewers (continued)**
>
> **3. The benefit of using sentence embeddings as attribute representation**
>
> We agree that language-input models are competitive with or occasionally outperforming the non-language counterparts (as observed by reviewer Ry2j), but such improvements are not significant. One possible reason for this might be the increased number of parameters (recall the sentence embedding is a 768-d vector, which is much larger than the one-hot representation).
> We would like to note that, as the first work on relative behavioral attributes, the primary goal of this work is to discuss and verify possible computational frameworks that can realize the functionality of behavioral attributes across multiple domains. We understand that the advantage of using sentence embedding is the possibility of generalizing to unseen concepts that may be composed of known concepts. We can probably verify this in a more open-world experiment setup (e.g., a table-top manipulator that can accomplish a variety of tasks and can take any language instruction from the users). While this is beyond the scope of the current paper,  we will definitely explore this in our follow-up works.
>
> **4. Note**
>
> We corrected an error in the PbRL results in Table 2 in the Appendix. It does not always have a success rate of 0, but it is still significantly worse than our method. This does not change the conclusion, but for the purpose of integrity, we are reporting it here. We also carefully checked the results in all other experiments, they are all valid and the same as what we reported in the previous manuscript.

---

### Decision · Program_Chairs · 2023-01-20

**Decision:**

Accept: poster

**Justification For Why Not Higher Score:**

This paper proposes a novel solution to an important problem, but the reviewers agree that the current experiments are somewhat limited in their impact. They show that the idea is viable but more work will be needed to scale it up and apply to real world problems.

**Justification For Why Not Lower Score:**

This paper proposes a novel solution to an important problem, and the experimental results support the viability of the proposed method. It will be a valuable contribution to the community and hence should be accepted to ICLR.

**Metareview: Summary, Strengths And Weaknesses:**

This paper proposes a novel approach of doing RL against a reward model trained from human feedback. The authors propose using a set of attributes to describe the quality of an RL trajectory, and human feedback is collected against these attributes - i.e. does the trajectory need to be improved in any of the dimensions described by the attributes. This makes human feedback on each trajectory more precise and more informative, which leads to better performance.

The idea is novel and general with important practical applications. Saying this, the current work is seen by the reviewers as a proof of principle for the idea, and more work will be needed in the future to generalise this to the real world applications.

**Note From Pc:**

if the above contains the word "oral" or "spotlight" please see: "oral" presentation means -> notable-top-5% and "spotlight" means -> notable-top-25%. As stated in our emails, we are disassociating presentation type from AC recommendations